

# Can secondary contact following range expansion be distinguished from barriers to gene flow?

Johanna Bertl[1,2], Harald Ringbauer[3,4] and Michael G.B. Blum[5]

[1] Department of Molecular Medicine, Aarhus University, Aarhus, Denmark
[2] Vienna Graduate School of Population Genetics, Vetmeduni Vienna, Vienna, Austria
[3] Department of Human Genetics, University of Chicago, Chicago, IL, USA
[4] Institute of Science and Technology Austria, Klosterneuburg, Austria
[5] Laboratoire TIMC-IMAG, UMR 5525, Université Grenoble Alpes, CNRS, Grenoble, France

## ABSTRACT

Secondary contact is the reestablishment of gene flow between sister populations that have diverged. For instance, at the end of the Quaternary glaciations in Europe, secondary contact occurred during the northward expansion of the populations which had found refugia in the southern peninsulas. With the advent of multi-locus markers, secondary contact can be investigated using various molecular signatures including gradients of allele frequency, admixture clines, and local increase of genetic differentiation. We use coalescent simulations to investigate if molecular data provide enough information to distinguish between secondary contact following range expansion and an alternative evolutionary scenario consisting of a barrier to gene flow in an isolation-by-distance model. We find that an excess of linkage disequilibrium and of genetic diversity at the suture zone is a unique signature of secondary contact. We also find that the directionality index $\psi$, which was proposed to study range expansion, is informative to distinguish between the two hypotheses. However, although evidence for secondary contact is usually conveyed by statistics related to admixture coefficients, we find that they can be confounded by isolation-by-distance. We recommend to account for the spatial repartition of individuals when investigating secondary contact in order to better reflect the complex spatio-temporal evolution of populations and species.

## INTRODUCTION

Hybrid zones are narrow regions in which genetically distinct populations meet, mate, and produce hybrids (*Barton & Hewitt, 1985*). Hybrid zones induced by secondary contact have often been observed in connection with the Quaternary glaciations (*Hewitt, 2000*). For instance, molecular markers suggest that the southern peninsulas of Europe were major ice age refugia of the European biota and that secondary contact occurred during the northward expansion which followed the last glacial maximum (*Taberlet et al., 1998*; *Hewitt, 1999*). With the advent of multi-locus molecular markers such as microsatellite

Corresponding author
Johanna Bertl,
johanna.bertl@clin.au.dk

or SNP data, hybrid zones can be investigated using various molecular signatures including gradients of allele frequency, admixture clines, and local increase of genetic differentiation (*Nielsen et al., 2003*; *Adams, Lindmeier & Duvernell, 2006*; *Strand et al., 2012*; *Bermond et al., 2012*). Molecular or morphological clinal patterns provide evidence for secondary contact in various plant and animal species such as *Arabidopsis thaliana* (*Huber et al., 2014*), *Silene vulgaris* (*Keller & Taylor, 2010*), the grasshopper *Oedaleus decorus* (*Kindler, Arlettaz & Heckel, 2012*), the European hare *Lepus europaeus* (*Antoniou et al., 2013*) or the parrotbill bird *Paradoxornis webbianus* (*Qu et al., 2012*) to name just a few examples.

However, typical molecular signatures of secondary contact zones can also occur under other evolutionary scenarios. For instance, admixture clines can be observed under pure isolation-by-distance models where nearby populations are connected through gene flow (*Engelhardt & Stephens, 2010*). Additionally, an increase of genetic differentiation can occur in isolation-by-distance models when there are barriers to dispersal (*Barton & Bengtsson, 1986*; *Riley et al., 2006*). With the advent of landscape genetics, the search for barriers to gene flow has attracted considerable attention (*Manel et al., 2003*; *Storfer et al., 2010*). Although secondary contact zones can occur at barriers to gene flow, the two models convey different evolutionary paradigms. Models of barriers to gene flow are usually based on isolation-by-distance settings where neighboring populations are connected through dispersal (*Nagylaki, 1988*; *Safner et al., 2011*; *Blair et al., 2012*). Around the barrier to gene flow, dispersal is lowered because of geographical or anthropogenic obstacles (*Riley et al., 2006*; *Zalewski et al., 2009*). By contrast, models of secondary contact include an initial phase of evolutionary divergence between two populations or between two sets of populations. The phase of evolutionary divergence is followed by a phase of gene flow between the two divergent units at the secondary contact zone (*Murray & Hare, 2006*; *Durand et al., 2009*). The fact that patterns of genetic differentiation can be attributed to different demographic factors is a recurrent problem when using molecular markers. Genetic structure may represent past or contemporary processes and it is notoriously difficult to disentangle between the two possible explanations (*Epps & Keyghobadi, 2015*).

Here, we use coalescent simulations to investigate to what extent molecular data provide information to distinguish secondary contact following range expansion from barriers to gene flow. We consider neutral simulations only and the barrier to gene flow is modeled by a reduced migration rate. Models of secondary contact of neutral markers are non-equilibrium models that converge to a migration-drift equilibrium (*Endler, 1977*; *Bierne, Gagnaire & David, 2013*), whereas locally adaptive loci would rather converge to a selection-migration balance (*Barton, 1979*). For both evolutionary scenarios, we simulate a one-dimensional stepping-stone model as shown in Figs. 1 and 2.

For comparing the molecular signal left by the two distinct scenarios, we consider statistical measures that have been developed to provide evidence for different demographic processes. The first set of summary statistics, which is used to detect hybrid zones, contains measures of individual admixture coefficients between the parental source populations (*Nielsen et al., 2003*; *Durand et al., 2009*), and a measure of linkage

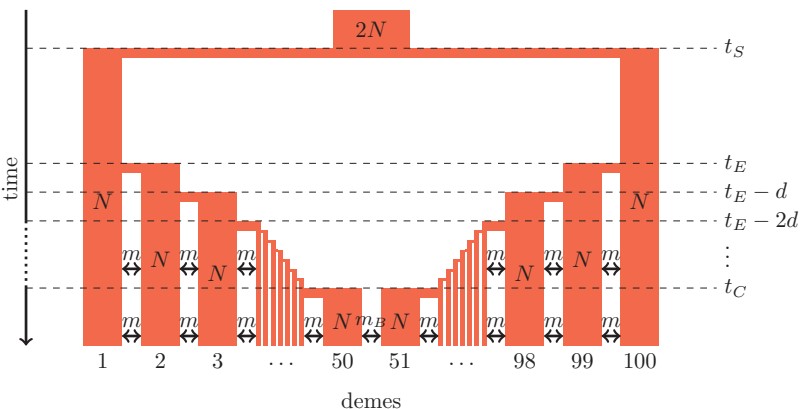

**Figure 1 Secondary contact model in a one-dimensional nearest-neighbor stepping-stone environment with 100 demes.** Parameters: $N$, deme size; $m$, scaled migration rate; $m_B$, scaled migration rate at the barrier ($m_B \leq m$); $t_S$, time of the population split; $t_E$, time when the expansion starts; $d$, time between two expansion steps; $t_C$, time of secondary contact.

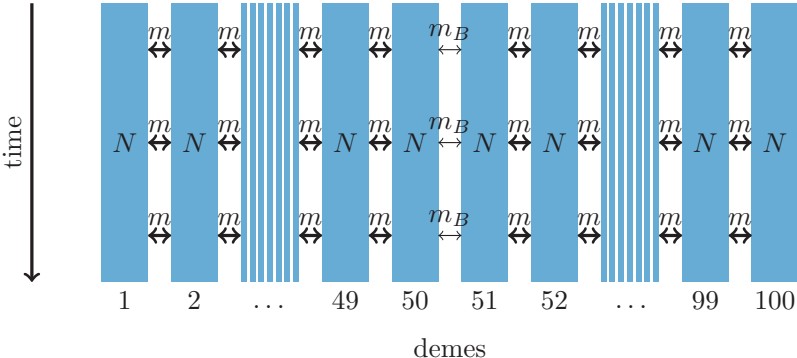

**Figure 2 One-dimensional model of a barrier to gene flow in a nearest-neighbor stepping-stone environment with 100 demes.** Parameters: $N$, deme size; $m$, scaled migration rate; $m_B$, scaled migration rate at the barrier ($m_B \leq m$).

disequilibrium (LD) as an increase of LD is expected in admixed populations (*McVean, 2002*).

The second set of summary statistics contains measures to detect range expansion because secondary contact is frequently induced by geographical expansions of the ancestral populations (*Hewitt, 2000*). We consider the directionality index $\psi$ as it is sensitive to the occurrence of recent range expansion and it should distinguish between equilibrium and non-equilibrium processes (*Peter & Slatkin, 2013*). The properties of the directionality index have not been studied yet when there are introgressive events. Furthermore, we include genetic diversity which has been shown to decrease along the expansion direction (*Austerlitz et al., 1997*).

The last set of summary statistics pertains to isolation-by-distance and barriers to gene flow. First, we include the decay of correlation between allele frequencies as a function of distance as it provides evidence of isolation-by-distance (*Hardy & Vekemans, 1999*). To detect barriers to gene flow, many numerical methods such as "wombling" identify
zones of sharp changes in allele frequencies (*Barbujani & Sokal, 1990*; *Manel et al., 2003*). Here, we use local $F_{ST}$ defined as $F_{ST}$ per unit of spatial distance for this purpose. Local $F_{ST}$ can be provided with georeferenced data by the software *LocalDiff* and we expect them to be larger around the barrier to gene flow (*Duforet-Frebourg & Blum, 2014*). The development of the software *LocalDiff* questioned the possibility of distinguishing between the two evolutionary scenarios under consideration. When studying patterns of differentiation in various alpine plants with *LocalDiff*, we found a region of larger local genetic differentiation in the Italian Aosta valley that was shared across alpine plant species. Both secondary contact zone following postglacial expansions or a barrier to gene flow in a equilibrium stepping-stone process were putative explanations (*Duforet-Frebourg & Blum, 2014*).

## METHODS

### Simulation models

We consider secondary contact in a one-dimensional nearest-neighbor stepping-stone model consisting of 100 demes (Fig. 1). Range expansion is modeled as a series of founder events with moderate bottlenecks. Time is given in coalescent units before present time, that is, in units of $4N$ generations where $N$ is the diploid population size per deme at present time. Accordingly, all parameters are scaled with $4N$.

**Phase 1 (ancestral population).** The ancestral population is a random-mating population of size $2N$. At time $t_S$, it splits in two populations of size $N$.

**Phase 2 (separate refugia).** From time $t_S$ to time $t_E$, the two populations are in separate refugia (demes 1 and 100, respectively), the population sizes are constant, and there is no gene flow.

**Phase 3 (expansion).** Starting at time $t_E$, both populations expand toward each other in the stepping-stone environment. At time points $t_E$, $t_E-d$, $t_E-2d$ etc., 10% of the individuals of the deme at the expansion front colonize a new deme. Instantaneously, the size of both demes increases to $N$ again and migration occurs at rate $m$ between neighboring demes.

**Phase 4 (secondary contact).** From $t_C = t_E-48d$ until the present time, a stepping-stone model with 100 demes of size $N$ is maintained with a migration rate $m$ among the neighboring demes and a migration rate $m_B \leq m$ between demes 50 and 51 where secondary contact occurs. If secondary contact occurs at a barrier, $m_B < m$, otherwise $m_B = m$.

As an alternative demographic model we investigate a nearest-neighbor stepping-stone model with a constant range of 100 demes and reduced gene flow in the center (Fig. 2). Again, the barrier to gene flow is modeled by a lower migration rate $m_B \leq m$ between demes 50 and 51.

DNA data of 20 haploid (or 10 diploid) individuals per deme is simulated with the coalescent simulator `ms` (*Hudson, 2002*). For each of them, we simulate 100 unlinked sequences consisting of 100,000 base pairs each. A sequence contains 100 SNPs and the scaled recombination rate within the sequence is 4.

In the secondary contact model we simulate data with parameters $t_S = 19$, $d = 1/8$ and different durations since secondary contact occurred (secondary contact is maintained from $t_C = 0, 1, \ldots, 5$ until present time). In both models the scaled migration rate between neighboring demes is $m = 20$. In the stepping-stone model we consider different barrier permeabilities ($m_B/m = 0.002, 0.01, 0.02, 0.1, 0.2, 1$; a value of 1 denotes no barrier). To provide means and standard errors of the summary statistics, each simulation is repeated 100 times.

The `ms` command lines and the simulated data are available in the Figshare repository *Bertl, Blum & Ringbauer (2018)*, https://doi.org/10.6084/m9.figshare.4986545.v2.

## Summary statistics

In the following, we describe the summary statistics we calculated to disentangle the two evolutionary scenarios.

### *Hybrid zone summary statistics*

**Admixture coefficient.** Based on the first principal component we compute an admixture coefficient for the pool of the five demes left of the barrier (*Paschou et al., 2007*). The pools of the five leftmost and five rightmost demes are used as proxies for the two source populations. The admixture coefficient is defined as the average relative location of individuals in the putative admixed population with respect to the two source populations on the axis of the first principal component (*Bryc et al., 2010*). Let $\bar{y}_i$ be the score of the first principal component, averaged over the individuals in population $i = l, r, a$ (left, right source population, admixed population). Then, we define the admixture coefficient

$$\frac{|\bar{y}_a - \bar{y}_l|}{|\bar{y}_r - \bar{y}_l|}.$$

It takes values between 0 and 1 and is proportional to the fraction of genetic material inherited from the right source population through admixture.

The principal component analysis is conducted with the R function `prcomp` (*R Core Team, 2012*).

**Linkage disequilibrium.** We average the squared correlation coefficient between 1,000 randomly drawn pairs of SNPs within the same sequence over all unlinked sequences. We compute LD for each deme.

### *Range expansion summary statistics*

**Directionality index $\psi$.** The directionality index $\psi$ has been developed to detect range expansion and infer its origin (*Peter & Slatkin, 2013*). Its basis is that populations further away from the origin of an expansion have experienced more genetic drift. The index $\psi_{i,j}$ is a pairwise measure between demes $i$ and $j$ that compares the average allele frequencies in the two demes: stronger drift yields higher differences in allele frequencies.

Denote the allele frequencies in deme $i$ by the vector $f^{(i)} = \left(f_1^{(i)}, \ldots, f_L^{(i)}\right)$, where $L$ is the total number of SNPs. Then, the directionality index for demes $i$ and $j$, from each of which a sample of size $M$ has been drawn, is defined as

$$\psi_{ij} = \frac{1}{LM} \sum_{l=1}^{L} \left( f_l^{(i)} - f_l^{(j)} \right).$$

Given that a range expansion has occurred, $\psi_{i,j}$ should be negative if deme $i$ is closer to the origin of the expansion than $j$, and positive otherwise. If $\psi_{i,j} \approx 0$, both demes should be equally close to the origin of the expansion, or no range expansion has occurred. We compute values of $\psi_{26,j}$ with $j = 27, \ldots, 50$.

**Genetic variability.** We measure genetic variability in each deme by averaging the number of pairwise nucleotide differences between all pairs of sequences, denoted by $\Delta$.

### Summary statistics for isolation-by-distance and barriers to gene flow

**Allele frequency correlogram.** The Pearson correlation between the allele frequencies of demes $i$ and $j$ is denoted by $r_{i,j}$ and defined as

$$r_{i,j} = \frac{\sum_{l=1}^{L} \left( f_l^{(i)} - \bar{f}^{(i)} \right) \left( f_l^{(j)} - \bar{f}^{(j)} \right)}{\sqrt{\sum_{l=1}^{L} \left( f_l^{(i)} - \bar{f}^{(i)} \right)^2 \sum_{l=1}^{L} \left( f_l^{(j)} - \bar{f}^{(j)} \right)^2}}.$$

We compute the correlogram $r_{26,j}$ for $j = 27, \ldots, 75$.

**Local $F_{ST}$.** Local values of $F_{ST}$ correspond to pairwise $F_{ST}$ between neighboring demes (*Duforet-Frebourg & Blum, 2014*). Here, we use Weir and Cockerham's estimator for multiple loci and random union of gametes (*Weir & Cockerham, 1984*, p. 1363).

## Theoretical context

The simulated scenarios are partly amenable to theory. In particular, continuous spatial diffusion models have proven to be powerful approximations to discrete stepping-stone models (*Wright, 1943*; *Nagylaki, 1978*). These converge quickly to their continuous diffusive counterparts, especially in one spatial dimension (*Nagylaki, 1988*; *Barton, 2008*; *Forien, 2017*). Recently, such continuous diffusion models have been applied to scenarios of barriers to gene flow (*Ringbauer et al., 2018*), as well as secondary contact (*Sedghifar et al., 2015*). Here, we summarize relevant findings and provide predictions for our simulations where available.

### Diffusion predictions

In the modern coalescence framework, the spatial distribution of ancestral lineages back in time is often modeled by a diffusion process (*Wilkins & Wakeley, 2002*). A central parameter is the variance of spatial displacement in one time unit, $\sigma^2$. In a stepping-stone model this variance is $\sigma^2 = 2m$ (*Nagylaki, 1988*), thus in our simulations $\sigma^2 = 40$.

Diffusion theory gives rise to several predictions which we can test our simulations against. For free diffusion, the probability that an ancestral lineage is spatially displaced by distance $\Delta x$ at time $t$ back is given by a Gaussian with mean 0 and variance $\sigma^2 t$. In case of secondary contact, the expected fraction of ancestry that traces back to the left side of the contact zone is the probability of finding the ancestral lineage to the left of the barrier at time of secondary contact $t_c$. Integrating over all possible positions to the left of a barrier yields a cumulative Gaussian $\Phi(l/(\sigma \sqrt{t_C}))$, where $l$ is the geographic distance from

the contact zone (*Sedghifar et al., 2015*). This model predicts that the extent of significantly admixed ancestry covers approximately $\pm 2\sigma\sqrt{t_C}$ around the point of initial contact, and that on average clines dissipate as a cumulative normal distribution.

Another prominent but less straightforward prediction based on a diffusive limit is that pairwise $F_{ST}$ increases approximately linearly with pairwise distance, with slope $1/(4N\sigma^2)$ in a homogenous linear habitat (*Rousset, 1997*).

### Linkage disequilibrium

A pulse of admixture produces LD, that is, non-random associations of markers. This admixture LD decays approximately as $\exp(-\rho t_C)$ in a panmictic population, where $\rho$ is the recombination rate between the two involved loci (*Chakraborty & Weiss, 1988*). In contrast, in spatially structured populations concordant admixture clines keep producing LD. *Barton (1982)* derived that in recombination-migration model, a common measure for LD, $D = \mathbb{E}(PQ) - \mathbb{E}(P)\mathbb{E}(Q)$, equilibrates to

$$D = p'q'\frac{\sigma^2}{\rho},$$

where $p'$ and $q'$ are the spatial slopes of alleles $P$ and $Q$. This formula is valid if the slopes of the clines stay approximately constant over the timescale set by recombination ($\approx 1/\rho$). This signal of admixture LD is typically much stronger than background LD produced by random associations due to drift (*Hill, 1981*; *Neel et al., 2013*). *Sedghifar et al. (2015)* used a diffusion model to predict patterns of admixture LD in a one-dimensional habitat with recent secondary contact, similar to the scenario simulated here. They derived formulas for patterns of covariance of ancestry, and deduced from them that admixture LD in a one-dimensional secondary contact model breaks down slower than $\exp(-\rho t_C)$, with a peak at the center of the contact zone.

### Genetic variability

Diversity decreases along an expanding wave of colonization (*Austerlitz et al., 1997*; *Edmonds, Lillie & Cavalli-Sforza, 2004*), at a rate which depends on the speed of advance and the shape of the wave front (*Hallatschek & Nelson, 2008*). In admixed zones after secondary contact diversity can be largely increased, since coalescence of lineages originating from different sides is pushed back to before the initial split. According to the diffusion model, the spatial extent of this area of significant admixture will be $\approx \pm 2\sigma\sqrt{t_C}$ around the contact zone. The increase of diversity will depend on the time of isolation of the two populations, and the population structure before expansion.

In contrast, a barrier is not expected to influence within-population diversity substantially. The invariance principle (*Nagylaki, 1998*) states that the (correctly weighted) mean within-deme coalescence time, and thus the mean within-deme diversity, is independent of the migration matrix. While a barrier can markedly influence the recent coalescence time distribution (*Barton, 2008*), it will not strongly influence mean coalescence times. Therefore, one expects a stable $\Delta$ across the range of demes in the barrier model.

### Pairwise differentiation

In case of a secondary contact pairwise differentiation, here measured by $F_{ST}$, will be initially increased across the barrier. However, $F_{ST}$ for neighboring demes equilibrates quickly, with rate $\approx m + 1/2N$ (*Slatkin & Barton, 1989*), and the initial effect of secondary contact will not persist long on small geographic scales.

On the other hand, a barrier can distort equilibrium patterns of identity by descent (*Nagylaki, 1988*), and therefore patterns of pairwise $F_{ST}$. A barrier will only have a significant effect if the barrier is strong enough to markedly influence the spread of ancestry ($m_B/(m\sigma^2) \ll 1$) (*Barton, 2008*; *Ringbauer et al., 2018*).

## RESULTS

To test our simulations, we first compare our results to theoretical predictions from the diffusion model. Reassuringly, we find a close agreement. Average clines are accurately predicted by neutral diffusion expectations (Fig. S4). In the equilibrium case without a barrier, pairwise $F_{ST}$ increases approximately linearly with slope $1/(4N\sigma^2)$ (Fig. S5), as predicted by *Rousset (1997)*.

With the validity of the simulations confirmed, we compare the secondary contact model with no barrier at the contact zone ($m_B = m$) to the stepping-stone model with a barrier. We plot the summary statistics either as a function of time since secondary contact or of the intensity of gene flow across the barrier (Fig. 3; see Supplementary Section S5 for the underlying values). For summary statistics computed per deme (genetic diversity $\Delta$, LD) or per pair of neighboring demes ($F_{ST}$), we consider the pattern along the whole range of demes. Many important features are captured by the ratio between the values at the barrier or the suture zone, respectively, and the values to the left and right of it (see Figs. S1–S3 for examples of the pattern along the whole range of demes).

First, we consider the average admixture coefficient for the five populations that are located on the left-hand side of the barrier (demes 45–50). For the isolation-by-distance model with a barrier, these five populations are found to be admixed to an extent depending on the barrier permeability: when increasing the barrier permeability, admixture coefficients of individuals on the left-hand side of the barrier approach 50%. As expected, the populations are also found to be admixed in the secondary contact model without barrier (between 35% and 50%) except for the scenario where data is collected just before secondary contact occurs ($t_C = 0$).

The ratio between LD at the center (demes 49–52) and on both sides of the range (demes 24–27 and demes 74–77; demes closer to the edge of the range are skipped to avoid the edge effect; see Figs. S1–S3) shows that LD is homogeneous along the whole range of demes for different barrier permeabilities in the stepping-stone model with a slight increase at the barrier for low values of $m_B/m$. However, in the secondary contact model, LD is considerably increased in the secondary contact zone, at the scale of mixed ancestry predicted by diffusion theory, $\pm 2\sigma\sqrt{t_C}$ (Figs. S1 and S2). The excess of LD ranges from a more than twofold to an approximately 1.3 fold increase and decreases as time since secondary contact increases.

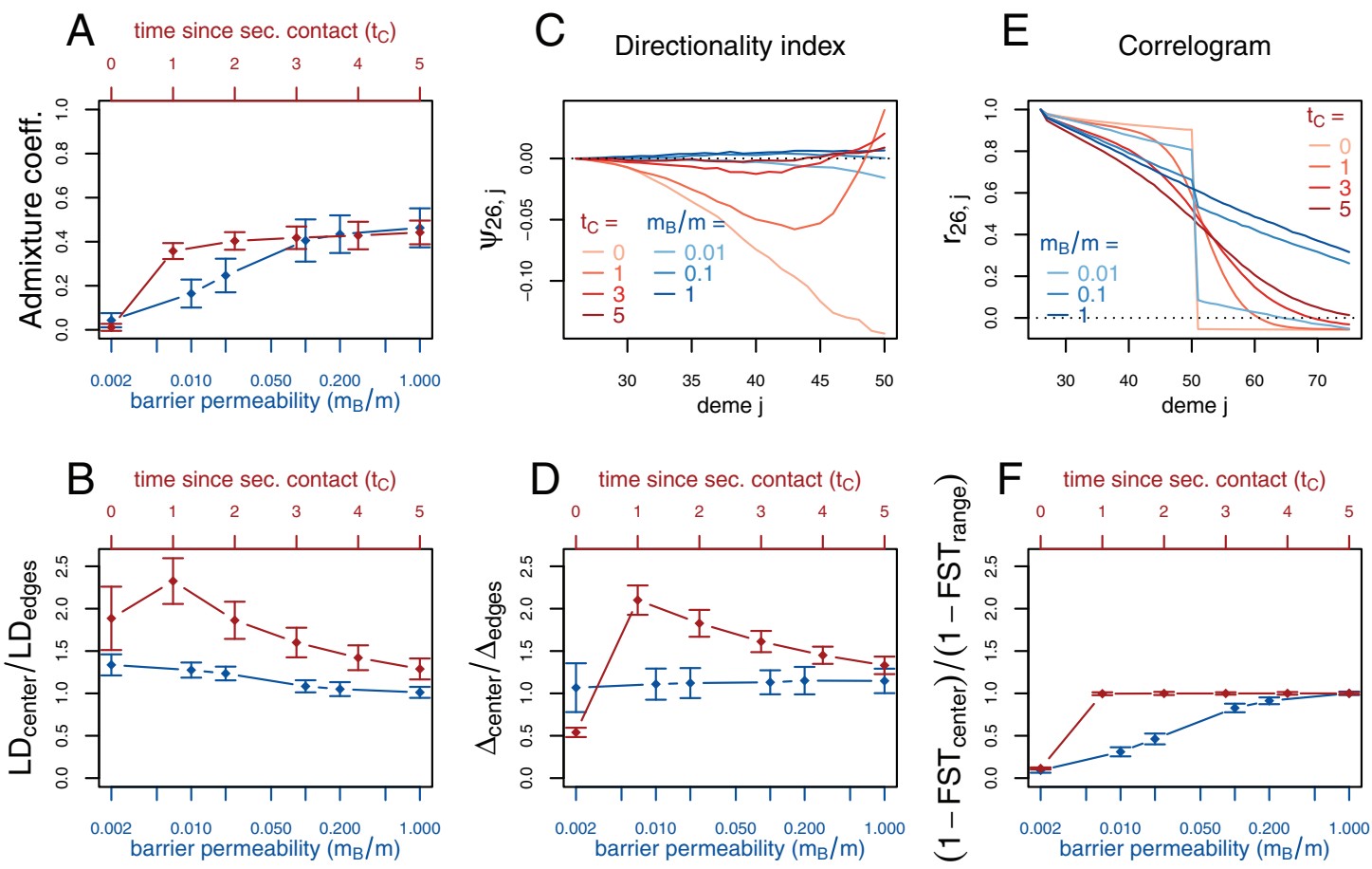

**Figure 3 Simulation results for the secondary contact model and the stepping-stone model with barrier. Admixture coefficient (A), Linkage disequilibrium (B), Directionality index $\psi$ (C), Genetic variability $\Delta$ (D), Allele frequency correlogram (E) and local $F_{ST}$ (F).** Red (lines and axes): secondary contact model with deme size $N = 1{,}000$, scaled migration rate $m = 20$ (constant migration rate at secondary contact zone, $m_B = m$), time of the population split $t_S = 19$, time since secondary contact $t_C = 0, 1, \ldots, 5$, time between two expansion steps $d = 1/8$. Blue: stepping-stone model with barrier with migration rate $m = 20$ and barrier permeabilities $m_B/m = 0.002, 0.01, 0.02, 0.1, 0.2, 1$; a value of 1 denotes no barrier. The barrier permeability $m_B/m$ is plotted on a logarithmic scale. The dots denote the mean and the error bars $\pm 2$ standard errors, estimated from 100 replicates of the simulations. For $\Delta$ and LD, the subscript *center* denotes the mean over demes 49–52 and *edges* over demes 24–27 and 74–77. The admixture coefficient is computed for the five demes to the left of the contact zone, demes 46–50. For the $F_{ST}$, *center* denotes demes 50 and 51 and *range* the mean over the neighboring demes in 26–74 except demes 50 and 51. (For these statistics, the edges of the range are dismissed because of the edge-effect in the steppingstone model.) For the allele frequency correlogram and the $\psi$ statistic, only the mean is plotted.

Apart from random fluctuations, the directionality index $\psi$ is constant for the stepping-stone model with constant migration rate ($m_B/m = 1$) as well as for old secondary contact ($t_C = 5$). More recent secondary contact results in a U-shaped pattern. The pairwise statistics $\psi_{26, i}, i = 27, \ldots, 50$ first decreases as expected when moving away from the origin of the expansion, but increases again toward the location of secondary contact. For the barrier model with a moderate or strong barrier ($m_B/m \geq 0.1$), $\psi_{26, i}$ remains constant for most of the range, but decreases slightly close to the barrier.

In the stepping-stone simulations the number of pairwise differences ($\Delta$) stays approximately constant over the range of demes (apart from an edge effect) and is hardly affected by the barrier (see also Fig. S3), as expected under the invariance principle.

Conversely, in the secondary contact model $\Delta$ increases in the suture zone, again at the spatial scale predicted by diffusion theory (Figs. S1 and S2). Only when secondary contact has not occurred yet ($t_C = 0$), the statistic $\Delta$ captures the effect of range expansion, and decreases when moving away from the origin.

When considering the decay of allele frequency correlation as a function of distance, we find a sharp decrease around the suture zone or around the barrier, respectively. In the barrier model, the correlation decreases linearly with distance and drops sharply at the barrier. In the secondary contact model we observe a more sigmoid shape. For older secondary contact, the sigmoid decay converges toward the linear decay of the equilibrium stepping-stone model predicted by *Rousset (1997)*.

Pairwise $F_{ST}$ between neighboring demes is increased at the barrier to gene flow; the less permeable the barrier, the larger the ratio of $F_{ST}$ at the barrier compared to the rest of the range. In the secondary contact model, local $F_{ST}$ is increased at the center when measured just before secondary contact ($t_C = 0$), but it remains constant along the range of demes when secondary contact is already established ($t_C \geq 1$).

To assess the robustness of these results, we performed simulations of less extreme scenarios. Also with more moderate founder events, a lower expansion speed and higher migration rate between demes, we find that the pattern of LD, genetic diversity and the directionality index remain distinctive summary statistics (Supplementary Section S3). However, we also observe that the footprint of secondary contact is more difficult to detect for very slow expansions or high migration rates between neighboring demes (Figs. S7 and S8). But even with these parameter settings, the directionality index $\psi$ remains a discriminant statistic.

If secondary contact occurs at a barrier to gene flow, the difficulty of detecting the secondary contact from molecular data increases. We consider additional simulations where secondary contact occurs in a region where gene flow is reduced by a factor of 10 ($m_B/m = 0.1$). In many respects, we see an intermediate pattern between the two previously considered scenarios, yet, genetic diversity and the directionality index $\psi$ still provide evidence for secondary contact and LD is even more increased at the contact zone (Fig. 4).

Smaller subsets of the data are used to study the impact of the number of loci on the variance of the summary statistics. Analyses of smaller genomes with 10 and 20 unlinked sequences instead of 100 are presented in Figs. S9 and S10, respectively. The confidence intervals of all summary statistics are considerably larger, while the spatial pattern of the directionality index and the correlogram is preserved and visible, even through larger random fluctuations. The summary statistic least affected by the data reduction is local $F_{ST}$. LD and genetic diversity $\Delta$ provide a distinctive pattern for very recent secondary contact ($t_C = 1$) when the data consists of at least 20 loci, but not for 10 loci.

We also consider other sampling schemes where the amount of sampled data is reduced. When sampling four genomes per deme instead of 20, our results remain unchanged (Fig. S11). When sampling only from every fifth deme instead of every deme, they provide similar results for all but one summary statistic (Fig. S12). When reducing

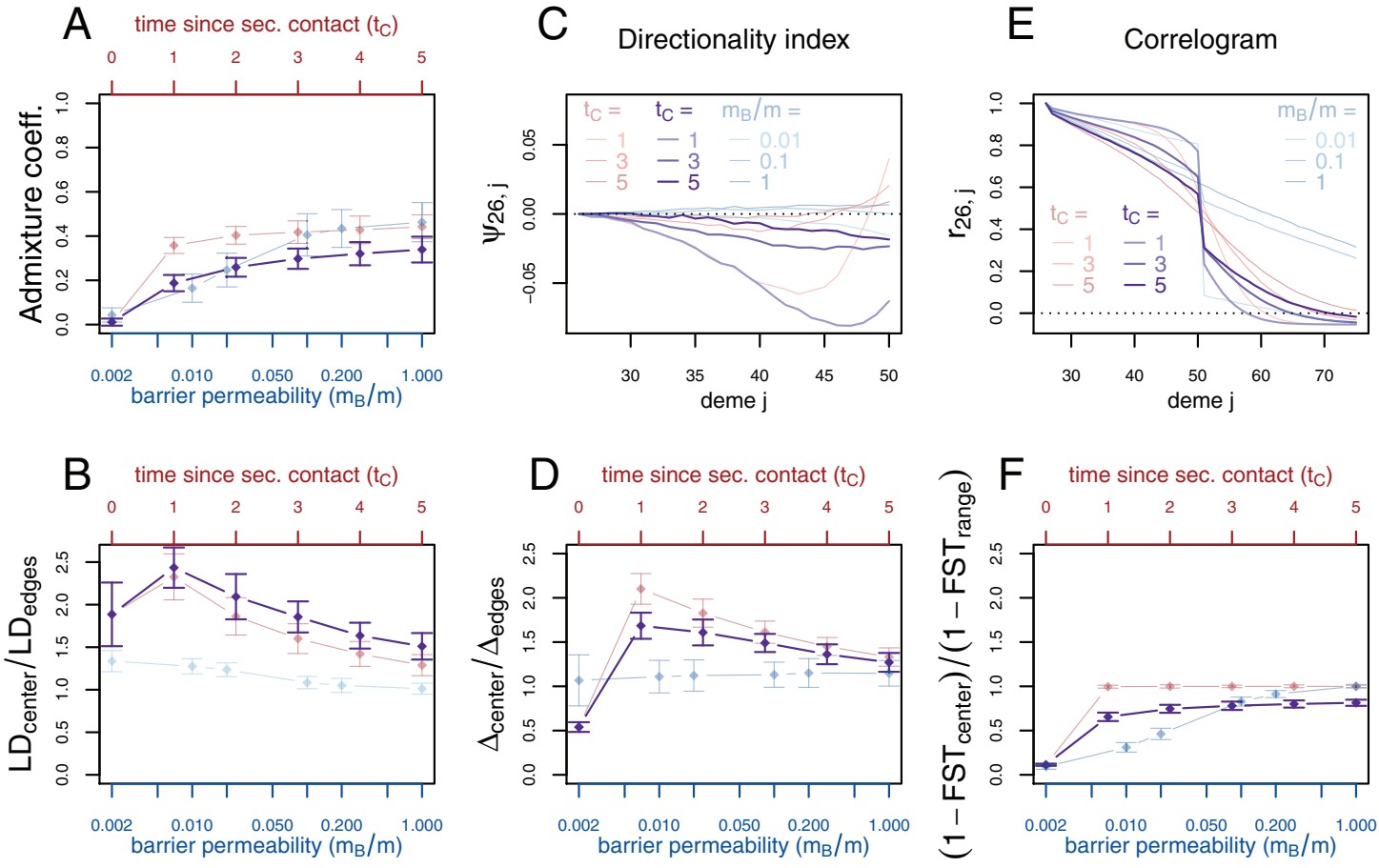

**Figure 4 Simulation results for the secondary contact model with barrier.** Admixture coefficient (A), Linkage disequilibrium (B), Directionality index ψ (C), Genetic variability Δ (D), Allele frequency correlogram (E) and local $F_{ST}$ (F). Black: secondary contact model with a moderate barrier to gene flow at the secondary contact zone ($m_B = 2$, $m = 20$). The remaining parameters are the same as for the secondary contact model in Fig. 3. For comparison, the secondary contact model without barrier (faint red) and the stepping-stone model (faint blue) are plotted again as in Fig. 3.

spatial sampling, it becomes more difficult to detect the spatial pattern of the directionality index ψ.

## DISCUSSION

We find that admixture coefficients alone do not provide sufficient evidence for secondary contact in the presence of isolation-by-distance. Some other summary statistics of genetic data such as local values of $F_{ST}$ or decay of correlation with distance were not more informative to identify the occurrence of secondary contact either.

By contrast, both an excess of LD and of genetic diversity at the suture zone are found to be unique signatures of secondary contact and also to be informative about the timing of secondary contact. In our simulations we observe an increase of these statistics in areas with mixed ancestry as predicted by the diffusion model ($\pm 2\sigma\sqrt{t_C}$). Several of these findings are well supported by previous theoretical considerations, as outlined above. In particular, admixture LD after secondary contact has been thoroughly studied before (*Sedghifar et al., 2015*). We do not detect such a marked increase of LD near

the barrier to gene flow in our stepping-stone model simulations, even for very strong barriers. It seems that the increase of differentiation across the barrier (which increases admixture LD) is roughly balanced by reduced migration across the barrier (which reduces admixture LD). This effect is not well documented in the literature, and warrants further theoretical investigation.

Although a peak of diversity can occur in glacial refugia (*Hewitt, 2000*), it has been observed before that the genetically most diverse populations were not located in southern Europe but at intermediate latitudes resulting from the admixture of divergent lineages that had expanded from separate refugia (*Petit et al., 2003*). Many hybrid zones have been formed by such postglacial secondary contact. In some cases, the divergence between hybridizing populations has evolved over very long timescales, and may have arisen already in primary contact. We still expect increased LD and admixture near the suture zones as unique signals of recent secondary contact then, since these signals are generated by allele frequency differences immediately before secondary contact. They do not depend on the evolutionary force that created this divergence.

We expect that excess of LD and of genetic diversity due to admixture would be similarly observed in a two-dimensional model. Qualitative patterns of these signals will be affected by the movement along the second dimension, but the underlying theoretical considerations due to admixture LD and increased diversity due to deeper coalescence time remain the same. The edge effects observed here, that is, reduced diversity and excess of LD because of shorter coalescence times (*Wilkins & Wakeley, 2002*), will be most pronounced at the four corners of the species range then (*Wilkins, 2004*).

We also find that the directionality index $\psi$ conveys a signature of secondary contact following expansion. Under range expansion, the $\psi$ statistic is a monotonous function of the distance from the origin of the expansion (*Peter & Slatkin, 2013*) (see also Fig. 3 for $t_C = 0$). When secondary contact follows range expansion, it has a distinctive U-shaped pattern (Fig. 3). The distinctive U-shaped pattern is found only for recent enough secondary contact ($t_C = 1$) but is robust to a wide range of bottleneck intensity and expansion speed values (Figs. S6 and S7). This statistic adds to the toolbox of population geneticists and provides a promising attempt to distinguish between equilibrium and non-equilibrium spatial processes.

The simulation setting was designed to mimic the evolutionary history of species that have undergone a population split during the Quaternary glaciations with subsequent expansion and secondary contact. Assuming a generation time of 1 year and 1,000 diploid organisms per deme, it includes the time frame of expansion and secondary contact after the last glacial maximum in Europe. Species that had spent the last glacial period in southern refugia started to expand northward around 16,000 years ago, and subsequently, many plants established a stable distribution around 6,000 years ago (*Hewitt, 1999*). We assume the ancestral population split up and started diverging 38,000 years ago ($t_S = 19$) and the onset of the expansion varies from 16,000 to 6,000 years ago and lasted 6,000 years. Finally, secondary contact is established on the range of 10,000 years ago ($t_C = 5$) to present time ($t_C = 0$; in this setting, both populations have expanded, but no gene-flow has occurred yet). Our simulations show that the molecular signal of secondary

contact vanishes after approximately 10,000 years. To apply our results to a specific organism, parameters like effective population size, time of divergence and expansion rate need to be calibrated.

In additional simulations, we found that the same summary statistics are distinctive for a wide range of parameter values, and also in datasets with fewer samples. However, reducing the number of loci increases the variance of the summary statistics considerably as it has been observed in coalescent models of single populations or isolation-migration models (*Felsenstein, 2006*; *Wang & Hey, 2010*).

Our findings are relevant when investigating modes of speciation using computational approaches (*Becquet & Przeworski, 2009*). Secondary contact following divergence without gene flow (allopatry) is often compared to models of speciation where species start diverging while exchanging migrants (sympatry or parapatry) (*Becquet & Przeworski, 2009*; *Duvaux et al., 2011*; *Roux et al., 2013*). The different frameworks to study speciation are based on isolation-and-migration models, which do not account for the spatial and potentially continuous repartition of individuals (*Pinho & Hey, 2010*). As shown in our simulation study, accounting for spatial processes provides additional information that can partly be caught with the $\psi$ directionality index, which has power to reveal evolutionary events such as secondary contact and range expansions.

The fact that isolation-by-distance affects the ascertainment of population structure has already been documented (*Novembre & Stephens, 2008*; *Frantz et al., 2009*). Accounting for space is a general recommendation that also stands when studying admixture between divergent populations of the same species (*Patterson et al., 2012*). Although isolation-by-distance is usually perceived as a confounding factor (*Meirmans, 2012*), the spatial sampling of individuals is in fact a chance to develop more powerful statistical approaches in evolutionary biology. Accounting for continuous populations should also be possible when performing simulations to choose the most probable scenario of speciation (*Duvaux et al., 2011*). Numerical simulators of genetic variation that account for the spatial repartitions of individuals are now available (*Ray et al., 2010*; *Kelleher, Barton & Etheridge, 2013*). We hope that these developments will encourage researchers to study speciation models that reflect the complex spatio-temporal dynamics of realistic species' evolutionary histories (*Alvarado-Serrano & Hickerson, 2015*).

## ACKNOWLEDGEMENTS

The computational results presented have been achieved using the Vienna Scientific Cluster (VSC) and the GenomeDK HPC cluster at Aarhus University.

### Funding

Johanna Bertl was supported by the Vienna Graduate School of Population Genetics (Austrian Science Fund (FWF): W1225-B20) and worked on this project while employed at the Department of Statistics and Operations Research, University of Vienna, Austria.

This article was developed in the framework of the Grenoble Alpes Data Institute, which is supported by the French National Research Agency under the "Investissments d'avenir" program (ANR-15-IDEX-02). The funders had no role in study design, data collection and analysis, decision to publish, or preparation of the manuscript.

### Grant Disclosures

The following grant information was disclosed by the authors:
Vienna Graduate School of Population Genetics (Austrian Science Fund (FWF)): W1225-B20.
Department of Statistics and Operations Research, University of Vienna, Austria.
Grenoble Alpes Data Institute, which is supported by the French National Research Agency under the "Investissments d'avenir" program: (ANR-15-IDEX-02).

### Competing Interests

The authors declare that they have no competing interests.

### Author Contributions

- Johanna Bertl conceived and designed the experiments, performed the experiments, analyzed the data, prepared figures and/or tables, authored or reviewed drafts of the paper, approved the final draft.
- Harald Ringbauer authored or reviewed drafts of the paper, approved the final draft, theoretical background.
- Michael G.B. Blum conceived and designed the experiments, authored or reviewed drafts of the paper, approved the final draft.

### Data Availability

   Bertl, Johanna; Blum, Michael; Ringbauer, Harald (2018): Simulations of secondary contact after range expansion and a barrier to gene flow. figshare. Fileset. https://doi.org/10.6084/m9.figshare.4986545.v2

### Supplemental Information

Supplemental information for this article can be found online at http://dx.doi.org/10.7717/peerj.5325#supplemental-information.

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
