# Peer review of "Can secondary contact following range expansion be distinguished from barriers to gene flow?"

_PeerJ, doi:10.7717/peerj.5325_

## Round 0.1 · original submission · Major Revisions

Dear Authors,

First please excuse me for taking so long to get back to you with the reviews. The two reviewers had diametrically divergent recommendations, detailed reviews and all came during the holidays. So my evaluations took longer than normal.

In my own opinion, as presented, the article has a number of major weaknesses.

1) Weak presentation of theoretical background and previous studies.
2) Many variables are not introduce – I suggest an appendix with this information so that the readers need not look though original literature
3) Be explicit in how simulations were carried out – again I suggest a supplement with scripts to run the simulations
4) Provide an assessment of the statistical power of these tests – how many loci, frequencies, geographic scale. Basically some directions to readers for how to apply your tests to their data.

There are number of other issues and specific points that need to be addressed, but both of the reviewers comment quite extensively on them already.

Based on these points and the evaluations of both reviewers, I recommend major revision.

I look forward to seeing a revised version in the near future.

Sincerely,

Tomas Hrbek

·

Basic reporting

The writing is generally clear, and the recent literature is referenced well. However, this issue goes back to (at least) the 1980s, and though Endler & Barton and Hewitt are cited, the basic arguments there about how secondary contact can be distinguished from primary are not explained.

The simulation results are given as figures; it is not clear how far the tables of results will be made available.


Sokal's (PNAS) work on "wombling" as a way to detect coincident divergence at multiple loci should be cited.

Experimental design

See below

Validity of the findings

The results are based on coalescent simulations, scaled with deme size, N. It is not at all clear how they relate to actual populations. In particular, LD is measured between "unlinked" sequences - but this is not equivalent to sampling unlinked genes in an actual population, because the recombination rate scales with N in the coalescent framework. As stated, these "unlinked" sequences should be uncorrelated in the coalescent limit, but would not be in a real population.

There is no consideration of sample size - a fixed # of sequences is used. This makes it hard to draw conclusions about statistcial power.

Bray et al's method is dubious because LD depends on allele frequencies, so that a correlation between dp and LD may occur without admixture.

Additional comments

Bertl & Blum use coalescent simulations of a one-dimensional chain of demes to investigate whether various statistics can distinguish secondary contact from isolation by distance in the presence of a barrier to gene flow. This is a long-standing and important question, which is now getting increased attention with the availability of extensive genetic data. However, the treatment is very limited, and is not related to exisiting theory. A thorough examiniation of these issues would be welcome, but this paper ius very far from that.

The various statistics are nit defined in the paper, so one has to go back to the original papers. Actually, these statistics are closely related - for example, Fst as a function of distance is another way to represent spatial autocorrleation of allele frequencies. These relations would be clearer if the definitions were set out using a common notation.

Most data come from two-dimensional populations over a very large range, relative to local dispersal. It is well known that both range size and dimension have strong effects on genetic differentiation. Thus, these results could be misleading if compared with actiual data.

The effects of isolation by distance and of a barrier to gene flow in one and two dimensions are well-known theoretically, and the simulation results need to be compared with these - if only as a check that the simulations are correct. In fatc, comparison with theory would allow a much deeper understanding of how these statistics depend on the parameters than these limited simulations. Indeed, it woiuld become apparent that dimension and range size are curcial, as noted above.

Secondary contact generates a transient local increase in diversity and in LD, though a barrier will also generate a similar effect, albeit generally weaker. A basic starting point should be to explain the theoretcial expectation for how these quantities behave following secondary contact or in a stationary IBD model, and how this is expected to be reflected in the various statistics.

It needs to be made clear at the outset that the paper only deals with neutral processes, and that the barrier to gene flow is a purely physical barrier. GIven that there is much discussion of the effects of genetic barriers, there is scope for confusion here.

·

Basic reporting

I think this is a solid paper, clearly written and self-contained that would be of great interest to population geneticists.

I thank the authors for their effort to provide the simulated data (ms commands, simulation outputs) together with a detailed README file, both accessible on the Figshare repository. However, there is no explicit note in the main text indicating that the raw data is being deposited on Figshare, and the corresponding URL is also missing.

I have few minor comments:

• L87 – 89: Please give more details of why the development of LocalDiff questioned the possibility to discriminate between the two models.

• L463: Please replace “mutation rate” by “migration rate” in supplementary section 6.

• Fig1 and Fig2: Please consider giving the definition of the different parameters in the caption, so the readers can understand the figures without having to come back to the main text. Also, could you indicate directly on the schemes the different zones you define later (i.e., “centre”, “edges” and “range”)? It will help readers to visualize them.

• Fig3: Could you give in the caption the actual values used in the main simulations for all parameters (deme size, migration rate, time of split, expansion speed and bottleneck intensity)?

• Fig3 and FigS3: In the caption, does “Here, center denotes” refers to the FST plot?

• FigS6: Please consider changing purple with another, more distinct, color.

Experimental design

A key strength of this paper is in the rigour of the analysis, and I only have few suggestions.

2.1. I really appreciate that the authors explicitly accounted for space in the secondary contact model, both during range expansion and in the suture zone, as it is usually neglected in the speciation literature. However, I was somehow frustrated to see the analysis of a model of secondary contact with barrier to gene-flow relegated to the supplementary information (FigS6) and shortly commented in the discussion (L258 – 264). It has long been known that hybrid zone maintained by strong linkage disequilibrium among genetic incompatibilities are not stabilized geographically. They are expected to move until they reach a geographical barrier to dispersal (Barton 1979), a local region of low abundance (Nagylaki 1976) or an ecological boundary (Bierne et al. 2011). So secondary contacts usually coincide with regions of reduced gene-flow that maintain reproductive isolation. I think this should deserve more attention. Please consider clarifying the introduction starting from L45, and perhaps moving FigS6 in the main text.

2.2. It could be surprising to think about secondary contacts as converging to a migration-drift equilibrium (L45–48). The authors should distinguish between neutral loci and selected loci involved in reproductive isolation when saying that “secondary contact is a non-equilibrium situation that converges to a migration-drift equilibrium”. Neutral loci in a secondary contact model can quickly lose their initial association with the selected loci, and, indeed, converge to a migration-drift equilibrium. The genetic structure at neutral loci will depend on the geographic connectivity at a large scale, and can create isolation-by-distance patterns (Bierne 2013). On the contrary, the “speciation” loci are under selection-migration balance, and they are expected to decrease the effective migration rate of linked neutral loci (Barton 1979). It would be worth mentioning the distinction between selected and neutral loci in the introduction given that their molecular pattern are expected to be different.

2.3. Table 1 gives the results of the test for admixture (the null hypothesis being no admixture) in the case of secondary contact and isolation-by-distance. The authors have assessed the robustness of the test to varying times since secondary contact and varying intensities of the barrier to gene-flow. I would like to see how the power of the test (reject no admixture when there is admixture) and the false-positive rate (inferring admixture when there is isolation-by-distance) depends on varying the number of SNPs sampled? Also, what is the robustness of the test to varying level of migration rates in the secondary contact model? The authors may consider presenting these additional analyses in supplementary tables.

2.4. Since the primary goal of the paper is to discriminate between the two models based on a set of statistics, I think the authors should give some practical hints to readers who may be interested in applying the method to their study system. In particular, the directionality index seems promising.

Referring to lines 76–77, I would have liked to see how the U-shaped pattern of the directionality index in the case of secondary contact with expansion (Fig3) compares to a simple expansion model?

Following Peter & Slatkin (2013), could the authors perform a statistical test (the null hypothesis being no expansion) to determine the significance of the directionality index between demes? For example, by producing a null distribution based on random permutations of allele frequencies. How does the power of such test (reject isolation-by-distance when there is expansion), and the false-positive rate (inferring expansion when there is isolation-by-distance) depends on different bottleneck intensities and expansion speeds?

In a more practical side, I wonder to which extent the results presented in Fig3 (and in the corresponding supplementary figures) depend on the delimitation of the edges (demes 24–27 and 74–77) and of the centre (demes 49–52 or 50–51). This is obvious for patterns of linkage disequilibrium and genetic diversity for which different signatures are expected if only demes from 50 to 60 are sampled in the field (see FigS1). Could the authors comment on that?

Validity of the findings

The description of the results is clear and the authors did not over-interpret them. Basically, secondary contact can be easily distinguished from migration-drift balance only when the contact is sufficiently recent compared to the total divergence time between the two populations.

I would have moderated the conclusions made lines 248–250, as the directionality index has a distinctive pattern under secondary contact only when the contact is recent (Tc=1, see Fig3). Moreover, this depends entirely on having a series of bottleneck events during range expansion (though, it may be a common feature of population history).

Additional comments

In this manuscript, Bertl and Blum ask whether two spatially explicit models, namely a secondary contact following range expansion, and an isolation-by-distance model with a barrier to dispersal, can be distinguished based on a set of commonly used statistics. The authors address this question by simulations, mirroring the evolutionary history of species which started to diverge during the last Quaternary glaciation in different glacial refuges, and subsequently expanded their range until secondary contact.

Although many recent studies have been conducted to compare different modes of speciation, little has been done to explicitly account for range expansion during the re-colonization process and spatial structure in the suture zone. I particularly appreciate this novelty. The contribution of this paper is also to assess the power of a set of statistics in discriminating between a non-equilibrium secondary contact model and an isolation-by-distance model. I found interesting the use of the directionality index, which leaves a specific pattern in the secondary contact model.

I only have two main negative comments. First, I regret that the authors only considered in the supplemental information a model of secondary contact with a barrier to gene-flow. It is well known that hybrid zones often coincide with geographic barriers to dispersal (Hewitt, 2000), so it would have been worth considering this scenario in the main text. Second, I think it would be useful for readers who are interested in applying the directionality index to their study system to give them a way of testing for the presence of range expansion (as the authors did for the presence of admixture, see Table1).


References

Barton, N (1979). Gene flow past a cline. Heredity, 43 (3), 333–339.
Bierne N, Welch J, Loire E, Bonhomme F, and David, P (2011). The coupling hypothesis: why genome scans may fail to map local adaptation genes. Molecular Ecology, 20, 2044–2072.
Bierne, N, Gagnaire, P-A, and David, P (2013). The geography of introgression in a patchy environment and the thorn in the side of ecological speciation. Current Zoology, 59(1), 72–86.
Hewitt, G (2000). The genetic legacy of the Quaternary ice ages. Nature, 405 (6789), 907–913.
Nagylaki, T (1976). Clines with variable migration. Genetics, 85, 867–886.
Peter, B, and Slatkin, M (2013). Detecting range expansions from genetic data. Evolution, 67, 3274–3289.

---

## Round 0.2 · Major Revisions

Dear Authors,

I have now received two reviews from both the original referees. Again I have two very divergent reviews, a reject and a minor revision.

I understand that your study is a simulation and not a review of the mathematical theory of secondary contact. This is not what is being asked of you. The point rather is that there are straightforward predictions for primary divergence and secondary contact, and this should be taken into account in the analyses. So in the simulations one can look for these specific signatures, and not rely just on summary statistics, some of which may not be entirely appropriate.

If you are able to rework the paper to address these issues, I do not see why it should not be acceptable for publication.

I look forward to seeing a revised version in the near future.

Sincerely,
Tomas Hrbek

·

Basic reporting

The structure writing is generally clear, though the English could be improved. Some points are ambiguous (see below). The references are largely to recent "landscape genetics" plus some older reviews, but there is no engagement at all with the substantial theoretical literature.

Experimental design

The simulations are very limited, and in particular represent small one-dimensional populations. Patterns will be very different in large 2D populations - which indeed present a considerable computational challenge.

The simulations are only examined using various summary statistics; there is bio examination of the actual patterns driving these statistics, or checks that they match the relevant theory.

Validity of the findings

As explained below, these limited simulation results are likely to be misleading.

Additional comments

The authors say that this is a simulation paper, and are not willing to review the relevant theory. Sometimes, the theory is intractable, and so one can only simulate. However, in this case there are many straightforward theoretical predictions that give insight into how one might distinguish secondary contact from primary divergence. One cannot simply ignore it.

One such prediction is that LD is generated by a short-term balance between admixture and recombination. Yet, it is claimed here that LD can be used to distinguish primary from secondary contact. I suspect that this is just because neutral clines following secondary contact may be much sharper, relative to dispersal.; this will dissipate quickly, except near selected loci, with LD decaying roughly as 1/(rt). This explanation needs to be checked: is the signal coming from differences in cline width?

A central problem, which is not discussed at all, is that although many hybrid zones may have been formed by postglacial secondary contact, the divergence between hybridising populations has evolved over much longer timescales, and may have arisen in primary contact. Moreover, the appropriate timescale depends on the spatial scale: near to a barrier, patterns are likely due to recent isolation-by-distance, but fiurther away, reflect much more ancient divergence. This has been quantified by Nagylaki, and most recently by Barton (2008).

The paper compares various summary statistics under different simulated scenarios, but does not go beyond that, to see what signals are driving the statistics. These statistics rely on either allele frequency differences or LD; one needs to know how the different scenarios affect the underlying genotype frequencies, rather than relying just on some summary statistics. More fundamentally, the various statistics were devised to test specific models, and should not be used in different contexts.

115 - Scaling to 2N is more usual than to 4N. Actually, this scaling in problematic in a 1D population, and more important, a small cluster of demes in 1D is completely unrepresentative of a large 2D populatuiion, where coalescent timescales are far longer.

line 137 - If the loci are unlinked, =how can they be on the same chromosome? Is the model a genetic map length 4 with 100 non-recombining blocks each with 100 SNP? This is not at all clear.

142 - m is more usual to denote mgration rate - mu denotes mutation.

162 - This equation is not appropriate for a 1D cline when there is high migration between adjacent demes, and when there is continued gene flow, not a one-off admixture event.

250 "secondary contact" should not be contrasted with "steppong stome model"!

·

Basic reporting

'no comment'

Experimental design

'no comment'

Validity of the findings

'no comment'

Additional comments

Bertl and Blum provide an improved version of their manuscript.

They made substantial effort to better evaluate the robustness of their patterns to different sampling schemes (provided in the new section S3); and to assess the power of the admixture test under a range of parameter combinations (provided in new Tables S1-S3). I think these improvements will give useful directions to readers who want to apply the method to their data.

Overall, the methods and figures have been clarified. The authors introduce in more details the different statistics (though, calculation of the admixture coefficient must be made more explicit), and the data & scripts are made available (Figshare repository and Table S4).

I also appreciated that the authors now fully consider the scenario of a secondary contact with a barrier to gene flow in the main document; and they make explicit the difficulty to distinguish it from IBD in the presence of a barrier to gene flow.

Nevertheless, I am not totally satisfied by the answers relative to weaknesses of the theoretical background.
First, although the authors did precise the concepts of secondary contact and barrier to gene flow in the Introduction, I've found the explanations a bit confusing. In particular, the authors must consider rephrasing L44-46 (“not mutually exclusive” sounds peculiar) and L68-70. It is not clear what the authors consider to be “adaptive loci” on L70. An unconditional favourable allele will spread across the species range without being much delayed by a barrier to gene flow (Pialek & Barton 1997). Conversely, a locally adaptive locus will converge to a selection-migration balance.
Second, I agree with the other reviewer that a comparison of the simulation results with the theoretical expectations of IBD and a barrier to gene flow will be a safe check. At least, I would like to see some evidences that simulations are correct in the simplest model.

---

## Round 0.3 · Minor Revisions

Dear Authors,

Thank you for taking time to revise this MS and addressing the comments of both reviewers. The MS has improved substantially, addressing and resolving the weak points of the previous versions. My concerns at this point are the same as those of Christelle Fraïsse who re-reviewed the current MS, and that is that the Material and Methods have several subsections of theoretical predictions/considerations (eg. “Comparison to the diffusion theory”) and then report results right away. These results are not reported in results and are not discussed. I realize that these were the “sanity checks”, but I think it would be much more effective if these “sanity checks” were also discussed, thus providing in a more explicit way, additional support for the findings of this study.
Christelle Fraïsse has several additional comments that I would like you to address. Otherwise I am happy with your revisions, and I look forward for to the final version of this MS in the near future.

Sincerely,

Tomas Hrbek

·

Basic reporting

'no comment'

Experimental design

'no comment'

Validity of the findings

'no comment'

Additional comments

I am satisfied with this third round of revisions, and I acknowledge the effort made by the authors to provide a theoretical context to their work. In particular, the comparison of the simulation results to expectations of the diffusion theory under secondary contact (Figure S1, S2, S4) and a stepping-stone model (Figure S5) is reassuring, as the authors found an excellent agreement.

I have only one additional comment: at some places, I have the impression that the authors conduct a review of the theory instead of making direct connections with their simulation results. This impression is emphasized by the fact that all new theoretical context is reported in the M&M, while nothing is mentioned in the Results.
For example, the paragraph from lines 156 to 174 basically describes the good fit between simulations and quantitative expectations under the diffusion theory. It would be better included (or part of it) in the Results.
Similarly, the qualitative predictions for “Linkage disequilibrium” (lines 195 to 201), “Genetic variability” (lines 221 to 229) and “Allele frequency correlogram” (lines 242 to 245) may be better discussed in connection with the simulations in the Results.

As a minor point, I was also wondering why the analytical results on LD obtained by Sedghifar et al. (2015) were not use to directly compare the simulated spatial pattern of LD with its expectation? To which extent the model implemented by Sedghifar et al. differs from yours?


Minor comments
- Line 79: typo, “expansion because secondary contac[y] frequently induced by geographical...”

- Figure S8: “migration rate (μ)” should be replaced by “migration rate (m)”.

- Line 155 and 160: it is not clear what are the references for the formula “σ^2=2m” and “σ^2*t”.

- Line 199: this sentence seems a bit out of scope: “They demonstrated that patterns of admixture LD across space and linkage distances can be utilized to robustly infer the timing of secondary contact.”

- Line 241: extra bracket in ”Ringbauer et al. 2018 ))”.

---

## Round 0.4 · accepted · Accept

Dear Authors,

I am happy to accept your MS. You have done a commendable job addressing all the reviewer’s questions, and with these changes the MS is ready for publication.

Congratulations on a job well done. I am sure this MS and the methodology for studying secondary contact will be very useful for many researchers.

Sincerely,

Tomas Hrbek

#